# Circular Noncoding RNA hsa_circ_0003570 as a Prognostic Biomarker for Hepatocellular Carcinoma

**DOI:** 10.3390/genes13081484

**Published:** 2022-08-19

**Authors:** Se Young Jang, Gyeonghwa Kim, Won Young Tak, Young Oh Kweon, Yu Rim Lee, Young Seok Han, Ja Ryung Han, Jung Gil Park, Min Kyu Kang, Hye Won Lee, Won Kee Lee, Soo Young Park, Keun Hur

**Affiliations:** 1Department of Internal Medicine, School of Medicine, Kyungpook National University, Kyungpook National University Hospital, 130 Dongdeok-ro, Jung-gu, Daegu 41944, Korea; 2Department of Biochemistry and Cell Biology, Cell and Matrix Research Institute, School of Medicine, Kyungpook National University, 680 Gukchaebosang-ro, Jung-gu, Daegu 41944, Korea; 3Department of Surgery, Daegu Catholic University School of Medicine, 33, Duryugongwon-ro 17-gil, Nam-gu, Daegu 42472, Korea; 4Department of Internal Medicine, College of Medicine, Yeungnam University, 170 Hyonchung-ro, Nam-gu, Daegu 42415, Korea; 5Department of Pathology, Keimyung University School of Medicine, 1035 Dalgubeol-daero, Dalseo-gu, Daegu 42601, Korea; 6Department of Medical Informatics, School of Medicine, Kyungpook National University, 680 Gukchaebosang-ro, Jung-gu, Daegu 41944, Korea; 7BK21 FOUR KNU Convergence Educational Program of Biomedical Science for Creative Future Talents, Department of Biomedical Science, School of Medicine, Kyungpook National University, 680 Gukchaebosang-ro, Jung-gu, Daegu 41944, Korea

**Keywords:** circular RNA, epigenetics, hepatocellular carcinoma, survival, progression, biomarker, prognosis

## Abstract

Circular RNAs (circRNAs) are potential biomarkers owing to their stability, tissue specificity, and abundance. This study aimed to evaluate the clinical significance of hsa_circ_0003570 expression and to investigate its potential as a biomarker in hepatocellular carcinoma (HCC). We evaluated hsa_circ_0003570 expression in 121 HCC tissue samples, its association with clinicopathological characteristics, and overall and progression-free survival. Hsa_circ_0003570 expression was downregulated in HCC tissues. Low hsa_circ_0003570 expression was more common in tumors larger than 5 cm (odds ratio (OR), 6.369; 95% confidence interval (CI), 2.725–14.706; *p* < 0.001), vessel invasion (OR, 5.128; 95% CI, 2.288–11.494; *p* < 0.001); advanced tumor-node metastasis stage (III/IV; OR, 4.082; 95% CI, 1.866–8.929; *p* < 0.001); higher Barcelona Clinic Liver Cancer stage (B/C; OR, 3.215; 95% CI, 1.475–6.993; *p* = 0.003); and higher AFP (>200 ng/mL; OR, 2.475; 95% CI, 1.159–5.291; *p* = 0.018). High hsa_circ_0003570 expression was an independent prognostic factor for overall survival (hazard ratio (HR), 0.541; 95% confidence interval (CI), 0.327–0.894; *p* = 0.017) and progression-free survival (HR, 0.633; 95% CI, 0.402–0.997; *p* = 0.048). Hsa_circ_0003570 is a potential prognostic biomarker in patients with HCC, and further validation of hsa_circ_0003570 is needed.

## 1. Introduction

Circular RNAs (circRNAs) are single-stranded, covalently closed RNA molecules produced from pre-mRNAs through backsplicing. Advances in RNA sequencing and bioinformatics tools have enabled the discovery of various circRNAs and their functions [1]. CircRNAs have recently been identified as microRNA sponges [2], modulators of transcription [3], and protein-binding decoys or sponges. Moreover, circRNAs are known to function as hallmarks of cancer that can explain the transition of normal cells to cancerous cells [4]. CircRNAs are involved in various functions, including sustaining growth signaling, evading growth inhibitors, resisting apoptosis, uncontrolled replicative immortality, promoting angiogenesis, and activating invasion and metastasis [5]. Therefore, circRNAs have been shown to be related to tumorigenesis [6], epithelial–mesenchymal transition [7], and tumor progression in various cancers.

In addition to being cancer hallmarks, circRNAs have also been shown to be highly stable and can be found in exosomes, saliva, urine, and plasma [8,9]. Thus, circRNAs are considered good cancer biomarker candidates.

Hepatocellular carcinoma (HCC) is the most common type of primary liver cancer and the most common cause of death in people with cirrhosis [10]. The prognosis for HCC is poor because it tends to be diagnosed at an advanced stage. It is often asymptomatic until the tumor becomes large enough to tighten the liver capsule or compress adjacent organs or vessels with nerves, causing pain. The other cause of poor prognosis is that the state of the underlying liver disease limits treatment options and has an adverse effect on results, regardless of the HCC stage. Due to the immunosuppressive tumor microenvironment of HCC, new treatment targets and strategies have been proposed [11], and a new biomarker for predicting prognosis is necessary.

For investigating circRNAs as a biomarker in HCC, hsa_circ_00033570 was previously reported and analyzed in HCC tissues [12]. A previous study showed that the expression level of hsa_circ_0003570 was associated with clinicopathological characteristics. Therefore, we investigated the clinical significance and application of hsa_circ_0003570 as a biomarker in HCC.

## 2. Materials and Methods

### 2.1. Patients and Tissue Samples

This study included 162 patients with HCC who underwent a diagnostic biopsy or surgical resection at a single center between March 2015 and August 2016, and who have previously been studied [13]. We excluded patients who had previously been treated for HCC (*n* = 30) and those lost to follow-up (*n* = 9). Two patients were not checked for target circRNAs due to a shortage of tissue samples. Finally, 121 patients were included in the analysis. The median follow-up period was 24.5 months, ranging from 0.7 to 69.8 months.

The patients were monitored every three months using liver dynamic computed tomography (CT) or gadoxetic acid disodium–enhanced liver magnetic resonance imaging. HCC recurrence was recognized if a tumor exceeded 1 cm and showed contrast enhancement in the arterial phase and washout in the portal or delayed phase. Response Evaluation Criteria in Solid Tumors (version 1.1) was used to evaluate tumor response. We defined overall survival as the time between the date of initial HCC diagnosis, and either the date of death from any cause or the date of last contact with the patient. Progression-free survival was defined as the time between the initial date of HCC diagnosis, and either the first event of recurrence or progression or until death from any cause.

The tissue specimens for tumors and adjacent nontumor tissues were immediately stored at 4 °C for 24 h in RNAlater reagent (Ambion; Life Technologies, Carlsbad, CA, USA) and then stored at −80 °C. We collected patients’ clinicopathological data, including age, sex, etiology of liver disease, Child–Tucotte–Pugh (CTP) class, laboratory findings, α-fetoprotein (AFP) level, tumor size and number, presence of macrovascular invasion, and tumor stage. The tumor-node metastasis (TNM) stage, based on the criteria of the American Joint Committee on Cancer, 8th edition, and Barcelona Clinic Liver Cancer (BCLC) stage, was adopted. This study was approved by the institutional review board (KNUH-2014-04-056-001) and was conducted in accordance with the ethical guidelines of the 1975 Declaration of Helsinki. Written informed consent was obtained from all patients prior to sample collection.

### 2.2. Extraction of Total RNA and Synthesis of cDNA

Total RNA was extracted from the frozen tissues using QIAzol Lysis Reagent (Qiagen, Valencia, CA, USA) according to the manufacturer’s instructions. Total RNA samples were verified for concentration and purity using NanoPhotometer N60 (Implen NanoPhotometer, Westlake Village, CA, USA). Synthesis of cDNA was reverse transcribed using a High-Capacity cDNA Reverse Transcription Kit (Applied Biosystems, Foster City, CA, USA) following the manufacturer’s instructions.

### 2.3. Quantitative Real-Time Polymerase Chain Reaction

Quantitative real-time polymerase chain reaction (qRT-PCR) was performed with the SYBR Green PCR Master Mix (Applied Biosystems, Foster City, CA, USA) following the manufacturer’s instructions. For circular RNA expression analysis, the primers of hsa_circ_0003570 were designed, including a gap junction of circular RNA. The primers’ sequences of hsa_circ_0003570 were 5′- CAA GAT GGC ACA GCA GCA CAC GC -3′ (forward) and 5′- ATG CTG GTG CTC GGT TGG TC -3′. The primers’ sequences of lyceraldehyde 3-phosphate dehydrogenase (GAPDH), as a normalizer, were 5′- GGA AGG TGA AGG TCG GAG TC -3′ (forward) and 5′- GTT GAG GTC AAT GAA GGG GTC -3′. All of the primers were synthesized by Bionics (Seoul, Korea). qRT-PCR was performed in triplicate and amplification of hsa_circ_0003570 was confirmed via melt curve analysis. The relative expression results from the qRT-PCR were calculated with the 2^−ΔΔCt^ method.

### 2.4. Statistical Analysis

Categorical data are expressed as numbers (%), and numerical data are expressed as the mean and standard deviation for normally distributed data. For non-normally distributed data, the data are expressed as medians with interquartile ranges. To analyze the differences in hsa_circ_0003570 expression between the tumor and adjacent nontumor tissues, a paired *t*-test was used. To compare the clinicopathological characteristics between the two groups according to the hsa_circ_0003570 expression, we used the chi-square or Fisher’s exact probability test. Using the Kaplan–Meier method and log-rank test, we analyzed patient survival and compared survival between the groups. To identify the predictors of survival, we performed a logistic regression based on the Cox proportional hazards model. The statistical significance was set at *p* < 0.05. We conducted all the analyses using R statistical software 3.6.3 (the R foundation for Statistical Computing, Vienna, Austria; available at http://www.r-project.org, accessed on 5 March 2020), and the GraphPad Prism 6 program for Windows (GraphPad Software, La Jolla, CA, USA) was used to generate figures. 

## 3. Results

### 3.1. Baseline Characteristics of Hepatocellular Carcinoma Patients

Table 1 presents the baseline patient characteristics. Among the 121 patients, 19 (15.7%) underwent surgical resection and 46 (38.0%) underwent radiofrequency ablation as curative treatment. The other 11 patients (9.1%) underwent transarterial chemoembolization, 14 (11.6%) started sorafenib, and 31 (25.6%) received best supportive care as noncurative treatment. Seventy-three percent of the patients had viral hepatitis as an underlying disease.

### 3.2. Downregulation of hsa_circ_0003570 Expression in Hepatocellular Carcinoma Tissues

Figure 1 shows that the expression of hsa_circ_0003570 in HCC tissues was lower than that in noncancerous tissues (*p* = 0.001). 

### 3.3. Correlation between hsa_0003570 Expression and Clinicopathological Characteristics of Hepatocellular Carcinoma Patients

The differences in patient characteristics based on the expression levels of hsa_circ_0003570 are shown in Table 2. Based on the hsa_circ_0003570 expression, all patients were classified into high-expression (≥0.0005554) or low-expression (<0.0005554) groups. The cut-off value was determined using maximally selected rank statistics in the Maxstat R package. The optimal cut-offs were defined as the expression of hsa_circ_0003570 that best separated the two groups in terms of survival. Low hsa_circ_0003570 expression was associated with tumors larger than 5 cm (odds ratio [OR], 6.369; 95% confidence interval [CI], 2.725–14.706; *p* < 0.001); vessel invasion (Yes; OR, 5.128; 95% CI, 2.288–11.494; *p* < 0.001); advanced TNM stage (III/IV; OR, 4.082; 95% CI, 1.866–8.929; *p* < 0.001); higher BCLC stage (B/C; OR, 3.215; 95% CI, 1.475–6.993; *p* = 0.003); and higher AFP (>200 ng/mL; OR, 2.475; 95% CI, 1.159–5.291; *p* = 0.018).

### 3.4. Correlation between hsa_0003570 Expression and Survival of Hepatocellular Carcinoma Patients

The overall survival of the patients differed significantly according to the hsa_circ_0003570 expression (Figure 2a). The cumulative 1-, 2-, and 4-year overall survival rates were 40.0%, 31.1%, and 24.4%, respectively, in the low-expression group and 68.1%, 62.7%, and 40.0%, respectively, in the high-expression group. 

Table 3 shows the significant predictors of overall survival. Univariate analysis of prognostic factors for overall survival in patients with HCC demonstrated high hsa_circ_0003570 expression (hazard ratio [HR], 0.515; 95% CI, 0.331–0.801; *p* = 0.003); multiple tumors (HR, 2.408; 95% CI, 1.549–3.743; *p* < 0.001); AFP level > 200 ng/mL (HR, 2.938; 95% CI, 1.885–4.579; *p* < 0.001); poor CTP class (HR, 3.259; 95% CI, 1.881–5.648; *p* < 0.001); chronic hepatitis B (HR, 0.594; 95% CI, 0.383–0.922; *p* = 0.020); and curative treatment (HR, 0.141; 95% CI, 0.086–0.231; *p* < 0.001).

Multivariate analysis identified high hsa_circ_0003570 expression (HR, 0.541; 95% CI, 0.327–0.894; *p* = 0.017); poor CTP class (HR, 2.271; 95% CI, 1.122–4.595; *p* = 0.023); and curative treatment (HR, 0.171; 95% CI, 0.090–0.325; *p* < 0.001) as independent prognostic factors for overall survival.

### 3.5. Correlation between hsa_0003570 Expression and Progression-Free Survival of Hepatocellular Carcinoma Patients

Progression-free survival was significantly different between patients according to their hsa_circ_0003570 expression levels (Figure 2b). The cumulative 1-, 2-, and 4-year progression-free survival rates were 31.1%, 22.2%, and 15.6%, respectively, in the low-expression group, and 60.5%, 43.4%, and 26.3%, respectively, in the high-expression group.

Table 4 shows the significant predictors of progression-free survival. Univariate analysis of prognostic factors for progression-free survival in patients with HCC demonstrated high hsa_circ_0003570 expression (hazard ratio [HR], 0.580; 95% CI, 0.388–0.867; *p* = 0.008); multiple tumors (HR, 1.950; 95% CI, 1.316–2.889; *p* = 0.001); AFP level >200 ng/mL (HR, 2.767; 95% CI, 1.845–4.148; *p* < 0.001); poor CTP class (HR, 2.869; 95% CI, 1.676–4.911; *p* < 0.001); and curative treatment (HR, 0.186; 95% CI, 0.120–0.288; *p* < 0.001).

Multivariate analysis identified that high hsa_circ_0003570 expression (HR, 0.633; 95% CI, 0.402–0.997; *p* = 0.048); poor CTP class (HR, 2.163; 95% CI, 1.098–4.261; *p* = 0.026); and curative treatment (HR, 0.235; 95% CI, 0.133–0.416; *p* < 0.001) were independent prognostic factors for progression-free survival.

## 4. Discussion

CircRNAs are well-known potential biomarkers in cancer research because of their stability [14], tissue specificity [1], and abundance [15]. Numerous studies have reported the potential of these biomarkers in the diagnosis of various cancers [16]. CircRNAs can improve the performance of protein-based biomarkers, predict prognosis and treatment response, detect cancer early, and monitor recurrence. In HCC, some circRNAs have been reported to be associated with overall and recurrence-free survival after hepatectomy. For example, circTRIM33–12 [17], circSMARCA5 [18], and circADAMTS13 [19] act as the sponge of miRNAs to regulate HCC progression. In this study, we recruit not only early stage HCC cases treated by hepatectomy, but also advanced HCC cases beyond curative treatment, and showed the association between hsa_circ_0003570 and survival. 

We also discovered that hsa_circ_0003570 was downregulated in HCC compared to noncancerous human liver tissue. Its expression level was associated with various clinicopathological characteristics, particularly tumor size, vessel invasion, TNM stage, BCLC stage, and AFP level. Low expression of hsa_circ_0003570 was associated with poor overall and progression-free survival, among other clinical variables. 

Our results are consistent with those of a previous study [12]. Previously, hsa_circ_0003570 was reported to be downregulated in HCC cell lines and tissues. In that study, hsa_circ_0003570 was also associated with several clinicopathological characteristics, such as tumor size, tumor differentiation, microvascular invasion, BCLC stage, TNM stage, and AFP level.

The difference between the two studies is that we enrolled patients with advanced-stage HCC and considered the survival and progression of HCC according to the treatment modalities. To the best of our knowledge, this study is the first to discover an association between hsa_circ_0003570 and survival and progression in HCC patients. This raises the possibility of using hsa_circ_0003570 as a prognostic biomarker for HCC.

However, this study had some limitations. First, the retrospective nature of the study introduces a selection bias. We excluded 11 patients with HCC who were missing medical records due to loss to follow-up. Second, we could not acquire further pathological information, such as microvascular invasion or cell differentiation, owing to a needle biopsy performed in the advanced stage of HCC. Moreover, hsa_circ_0003570 cannot reflect the tumor heterogeneity of HCC. Therefore, it is necessary to recruit a larger number of HCC patients and validate hsa_circ_0003570 as a noninvasive biomarker using patient serum or urine. Third, we did not investigate the underlying mechanism of hsa_circ_0003570. Based on these results, we can only hypothesize that these circRNAs may act as a tumor suppressor.

## 5. Conclusions

In conclusion, we explored the clinical significance of hsa_circ_0003570 in patients with HCC. It was associated not only with the clinicopathological characteristics of HCC, but also with the survival and progression of HCC patients. Hsa_circ_0003570 can be a potential prognostic biomarker in patients with HCC, but further validation of hsa_circ_0003570 is needed in a future study.

## Figures and Tables

**Figure 1 genes-13-01484-f001:**
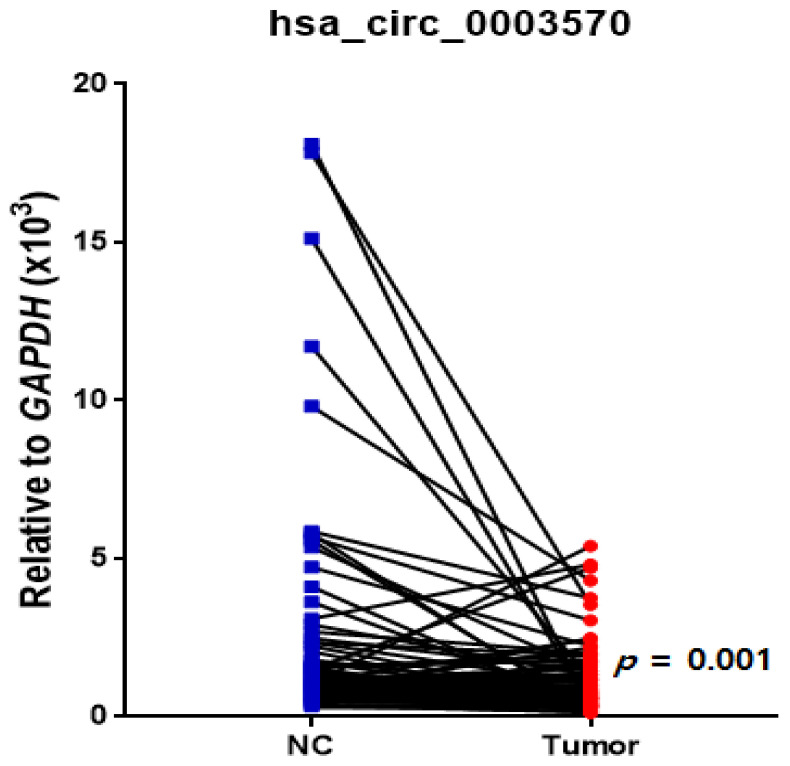
Dot plot of hsa_circ_0003570 expression in noncancerous (NC) tissue and cancer tissue in hepatocellular carcinoma. A paired *t*-test showed that hsa_circ_0003570 expression decreased in tumor tissue compared to that in the NC tissue (*p* = 0.001).

**Figure 2 genes-13-01484-f002:**
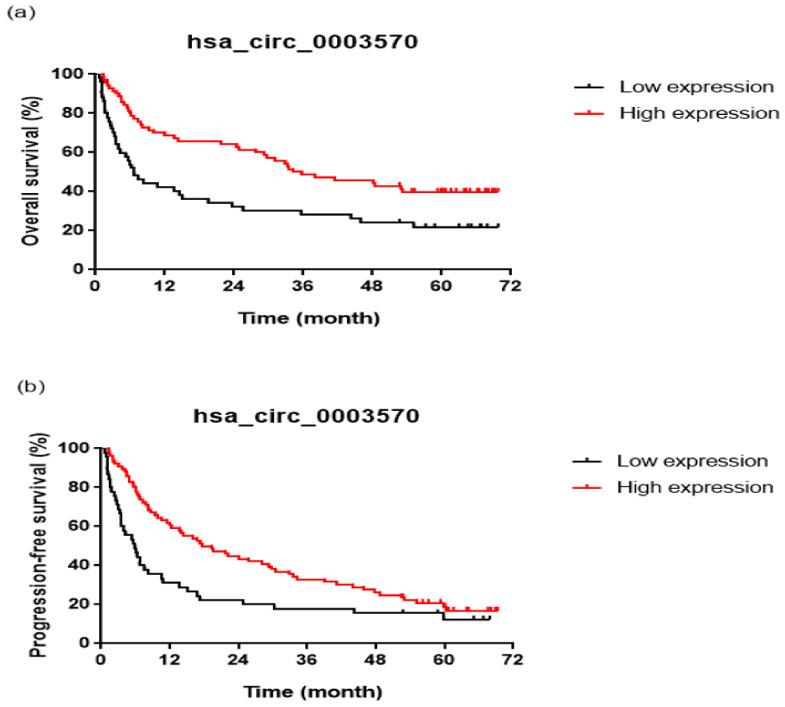
Survival curves according to the hsa_circ_0003570 expression level. Overall survival (**a**) (log-rank test, *p* = 0.002) and progression-free survival (**b**) (log-rank test, *p* = 0.007) rates were decreased in the low hsa_circ_0003570 expression level group.

**Table 1 genes-13-01484-t001:** Baseline characteristics of enrolled patients.

Clinical Characteristics	Total (*n* = 121)
Age (years)	60.8 ± 10.6
Sex	
Male	104 (86.0%)
Female	17 (14.0%)
Etiology	
HBV	74 (61.2%)
HCV	11 (9.1%)
Alcohol consumption	32 (26.4%)
HBV + HCV	2 (1.7%)
NASH	2 (1.7%)
Tumor number	
Single	67 (55.4%)
Multiple	54 (44.6%)
Size of tumor (cm)	
≤5	59 (48.8%)
>5	62 (51.2%)
Vessel invasion	
No	79 (65.3%)
Yes	42 (34.7%)
TNM stage	
I	51 (42.1%)
II	15 (12.4%)
III	13 (10.7%)
IV	42 (34.7%)
BCLC stage	
0/A	59 (48.8%)
B/C	62 (51.2%)
CTP class	
A	104 (86.0%)
B	17 (14.0%)
AST (U/L)	50.0 [29.0–74.0] *
ALT (U/L)	37.0 [25.0–53.0] *
Total bilirubin (mg/dL)	0.7 [0.5–1.1] *
Albumin (g/dL)	3.9 [3.5–4.2] *
Prothrombin time (s)	12.5 [11.8–13.2] *
AFP (ng/mL)	57.9 [7.5;2376.0] *

HBV—hepatitis B virus; HCV—hepatitis C virus; NASH—nonalcoholic steatohepatitis; TNM—tumor-node metastasis; BCLC—Barcelona Clinic Liver Cancer; CTP—Child–Turcotte–Pugh; AST—aspartate transaminase; ALT—alanine transaminase; AFP—α-fetoprotein; * median [25–75% interquartile range].

**Table 2 genes-13-01484-t002:** Clinical characteristics of patients according to hsa_circ_0003570 expression.

Clinical Characteristics	hsa_circ_0003570		
	Low *n* = 45)	High (*n* = 76)	*p*-Value
Age (years)			0.868
≤60	24 (53.3%)	38 (50.0%)	
>60	21 (46.7%)	38 (50.0%)	
Sex			0.656
Male	40 (88.9%)	64 (84.2%)	
Female	5 (11.1%)	12 (15.8%)	
Tumor number			0.360
Single	22 (48.9%)	45 (59.2%)	
Multiple	23 (51.1%)	31 (40.8%)	
Tumor size (cm)			< 0.001 *
≤5	10 (22.2%)	49 (64.5%)	
>5	35 (77.8%)	27 (35.5%)	
Vessel invasion			< 0.001 *
No	19 (42.2%)	60 (78.9%)	
Yes	26 (57.8%)	16 (21.1%)	
TNM stage			0.001 *
I/II	15 (33.3%)	51 (67.1%)	
III/IV	30 (66.7%)	25 (32.9%)	
BCLC stage			0.005 *
O/A	14 (31.1%)	45 (59.2%)	
B/C	31 (68.9%)	31 (40.8%)	
CTP classification			1.000
A	39 (86.7%)	65 (85.5%)	
B	6 (13.3%)	11 (14.5%)	
AFP (ng/mL)			0.030 *
≤200	21 (46.7%)	52 (68.4%)	
>200	24 (53.3%)	24 (31.6%)	
Chronic hepatitis B			0.994
No	18 (40.0%)	29 (38.2%)	
Yes	27 (60.0%)	47 (61.8%)	

TNM—tumor-node metastasis; BCLC—Barcelona Clinic Liver Cancer; CTP—Child–Turcotte–Pugh; AFP—α-fetoprotein. * *p* < 0.05.

**Table 3 genes-13-01484-t003:** Prognostic factors for overall survival in univariable and multivariable analyses.

**Factor**	**Univariable Analysis**		**Multivariable Analysis**	
	**Hazard Ratio (95% CI)**	***p-*Value**	**Hazard Ratio (95% CI)**	***p-*Value**
Hsa_circ_0003570 (high expression)	0.515 (0.331–0.801)	0.003 *	0.541 (0.327–0.894)	0.017 *
Age (>60 years)	1.100 (0.711–1.701)	0.669	1.397 (0.820–2.381)	0.219
Sex (male)	1.097 (0.594–2.026)	0.766	1.368 (0.679–2.756)	0.381
Tumor number (multiple)	2.408 (1.549–3.743)	<0.001 *	0.825 (0.463–1.470)	0.514
AFP (>200 ng/mL)	2.938 (1.885–4.579)	<0.001 *	1.457 (0.871–2.438)	0.152
CTP classification (B vs. A)	3.259 (1.881–5.648)	<0.001 *	2.271 (1.122–4.595)	0.023 *
Chronic hepatitis B	0.594 (0.383–0.922)	0.020 *	0.809 (0.486–1.347)	0.415
Curative treatment	0.141 (0.086–0.231)	<0.001 *	0.171 (0.090–0.325)	<0.001 *

CI—confidence interval; AFP—α-fetoprotein; CTP—Child–Turcotte–Pugh. * *p* < 0.05.

**Table 4 genes-13-01484-t004:** Prognostic factors for progression-free survival in univariable and multivariable analyses.

Factor	Univariable Analysis		Multivariable Analysis	
	Hazard Ratio (95% CI)	*p-*Value	Hazard Ratio (95% CI)	*p-*Value
Hsa_circ_0003570 (high expression)	0.580 (0.388–0.867)	0.008 *	0.633 (0.402–0.997)	0.048 *
Age (>60 years)	0.887 (0.600–1.311)	0.548	0.919 (0.575–1.467)	0.723
Sex (Male)	1.216 (0.713–2.076)	0.473	1.719 (0.945–3.125)	0.076
Tumor number (multiple)	1.950 (1.316–2.889)	0.001 *	0.754 (0.455–1.248)	0.272
AFP (>200 ng/mL)	2.767 (1.845–4.148)	<0.001 *	1.569 (0.963–2.557)	0.070
CTP classification (B vs. A)	2.869 (1.676–4.911)	<0.001 *	2.163 (1.098–4.261)	0.026 *
Chronic hepatitis B	0.766 (0.514–1.141)	0.190	0.967 (0.614–1.526)	0.887
Curative treatment	0.186 (0.120–0.288)	<0.001 *	0.235 (0.133–0.416)	<0.001 *

CI—confidence interval; AFP—α-fetoprotein; CTP—Child–Turcotte–Pugh. * *p* < 0.05.

## Data Availability

The datasets used or analyzed during the current study are available from the corresponding author on reasonable request.

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
