# Peer review of "Circular Noncoding RNA hsa_circ_0003570 as a Prognostic Biomarker for Hepatocellular Carcinoma"

_genes, 2022, doi:10.3390/genes13081484_

Round 1

Reviewer 1 Report

Jang et al found LncRNA (circ_0003570) was associated with clinicopathological characteristics, survival and progression in HCC patients. Therefore, circ_0003570 could be served as potential prognostics markers in HCC. However, there are some points which needed to be further fixed to support the conclusions.

1) The regulation mechanism of circ_0003570 in HCC should be investigated.

2) The p value should be shown in the figure 1 but not only in legend. 

3) How does this circulating LncRNA exist? Such as encapsulated by extracellular vesicles or under another form? 

4) Why does circ_0003570 secrete or express at a dysregulated level?

Author Response

1) The regulation mechanism of circ_0003570 in HCC should be investigated.

-> Thank you for your critical comment. This study focused on the clinical relevance and significance of hsa_circ_0003570 in predicting survival and progression rather than investigating its mechanism. We mentioned this as a limitation of our study in the discussion section, line 256.

-> “Third, we did not investigate the underlying mechanism of hsa_circ_0003570.”

2) The p value should be shown in the figure 1 but not only in legend. 

-> Thank you for your comment. We inserted p value on the figure 1.

3) How does this circulating LncRNA exist? Such as encapsulated by extracellular vesicles or under another form? 

-> Thank you for your critical comment. We investigated this circular RNA from tissues of HCC. Some of the circular RNAs might be circulating in the exosome (e.g., circMED27, PMID: 35024242), however, in this study, we focused on the tissue-based circular RNA.

4) Why does circ_0003570 secrete or express at a dysregulated level?

-> Thank you for your valuable comment. Circular RNA is known to be made during the process of RNA splicing, especially by backsplicing, however, the exact mechanism of circular RNA development has not been clearly understood until now. We just can suppose the answer based on this study and the previous study (reference 12), circ_0003570 might have a role associated with poor prognosis, for example, tumor proliferation or invasion.

Reviewer 2 Report

Se Young Jang et al. provide interesting data regarding HCC. Points to be considered: 1. I would suggest to slightly restructuring the manuscript as follows: P (Patient, population or problem) Who or what is the patient, population or problem in question? I (Intervention) What is the intervention (action or treatment) being considered? C (Comparison or control) What other interventions should be considered? O (Outcome or objective) What is the desired or expected outcome or objective? T (Time frame/treatment) 2. Did the authors check for hazard's proportionality before proceeding with multivariable analysis? 3. The authors could provide a little more consideration of genomic directed stratifications in clinical trial design and enrollments. The underlying message here is that more precision and individualized approaches need to be tested in well designed clinical trials – a challenge, but I would be interested in their perspective of how this might be done. 4. As is now well known, tumors grow and evolve through a constant crosstalk with the surrounding microenvironment, and emerging evidence indicates that angiogenesis and immunosuppression frequently occur simultaneously in response to this crosstalk. Accordingly, strategies combining anti-angiogenic therapy and immunotherapy seem to have the potential to tip the balance of the tumor microenvironment and improve treatment response. 5. In the frame of point 4 thinking this reviewer personally misses some insights in the introduction/ discussion sections regarding the fact that is one of most common cancers and the fourth leading cause of death worldwide. Commonly, HCC development occurs in a liver that is severely compromised by chronic injury or inflammation. Liver transplantation, hepatic resection, radiofrequency ablation (RFA), transcatheter arterial chemoembolization (TACE), and targeted therapies based on tyrosine protein kinase inhibitors are the most common treatments. The latter group have been used as the primary choice for a decade. However, tumor microenvironment in HCC is strongly immunosuppressive; thus, new treatment approaches for HCC remain necessary. The great expression of immune checkpoint molecules, such as programmed death-1 (PD-1), cytotoxic T-lymphocyte antigen 4 (CTLA-4), lymphocyte activating gene 3 protein (LAG-3), and mucin domain molecule 3 (TIM-3), on tumor and immune cells and the high levels of immunosuppressive cytokines induce T cell inhibition and represent one of the major mechanisms of HCC immune escape. Recently, immunotherapy based on the use of immune checkpoint inhibitors (ICIs), as single agents or in combination with kinase inhibitors, anti-angiogenic drugs, chemotherapeutic agents, and locoregional therapies, offers great promise in the treatment of HCC (please refer to PMID: 34065489 and expand)

Author Response

  1. I would suggest to slightly restructuring the manuscript as follows: P (Patient, population or problem) Who or what is the patient, population or problem in question? I (Intervention) What is the intervention (action or treatment) being considered? C (Comparison or control) What other interventions should be considered? O (Outcome or objective) What is the desired or expected outcome or objective? T (Time frame/treatment)

->Thank you for your valuable comment. The contents you mentioned are in a manuscript, and we summarized these contents according to your advice below.

P: Hepatocellular carcinoma patients who underwent diagnostic biopsy or surgical resection (as treatment-naïve) -> 2.1 Patients and tissue samples

I: To inspect the expression of hsa_circ_0003570 on HCC tissues -> 3.3. Correlation between hsa_0003570 expression and clinicopathological characteristics of hepatocellular carcinoma patients

C: To compare survival of progression according to the expression of hsa_circ_0003570 on HCC tissues -> 3.4. Correlation between hsa_0003570 expression and survival of hepatocellular carcinoma patients; 3.5. Correlation between hsa_0003570 expression and progression-free survival of hepatocellular carcinoma patients

O: To expect HCC patients with high expression of hsa_circ_0003570 will have better overall survival and progression-free survival -> Figure 2. Survival curves according to the hsa_circ_0003570 expression level.

T: Overall survival: the time between the date of initial HCC diagnosis and either the date of death from any cause or the date of last contact with the patient; Progression-free survival: the time between the initial date of HCC diagnosis and either the first event of recurrence or progression or until death from any cause. -> 2.1. Patients and tissue samples  

  1. Did the authors check for hazard's proportionality before proceeding with multivariable analysis?

-> Thank you for your critical comment. We checked for hazard's proportionality before proceeding with multivariable analysis. Because our study mainly focused on the expression of hsa_circ_0003570 and the clinical significance of HCC, we predicted the survival curve using variables other than expression of hsa_circ_0003570. By investigating log(-logS(t)) plots according to the expression of hsa_circ_0003570, the two plots were almost parallel (figure below), which could satisfy the assumption of Cox proportional hazard method (PHM).   

  1. The authors could provide a little more consideration of genomic directed stratifications in clinical trial design and enrollments. The underlying message here is that more precision and individualized approaches need to be tested in well designed clinical trials – a challenge, but I would be interested in their perspective of how this might be done.

->Thank you for your interesting comment. It will be important to validate the clinical significance of hsa_circ_0003570 and to investigate the potentiality of hsa_circ_0003570 as a prognostic biomarker. To validate this circular RNA, we will divide patients who underwent curative treatment by expression level of hsa_circ_0003570, then collect the clinical data of recurrence and survival. Statistical analysis will be needed to investigate the association between expression of hsa_circ_0003570 and prognoses (recurrence and survival).   

  1. As is now well known, tumors grow and evolve through a constant crosstalk with the surrounding microenvironment, and emerging evidence indicates that angiogenesis and immunosuppression frequently occur simultaneously in response to this crosstalk. Accordingly, strategies combining anti-angiogenic therapy and immunotherapy seem to have the potential to tip the balance of the tumor microenvironment and improve treatment response.

-> Thank you for your valuable comment. We totally agree with your opinion. We anticipate the clinical efficacy of anti-angiogenic therapy and immunotherapy in a real-world practice.

  1. In the frame of point 4 thinking this reviewer personally misses some insights in the introduction/ discussion sections regarding the fact that is one of most common cancers and the fourth leading cause of death worldwide. Commonly, HCC development occurs in a liver that is severely compromised by chronic injury or inflammation. Liver transplantation, hepatic resection, radiofrequency ablation (RFA), transcatheter arterial chemoembolization (TACE), and targeted therapies based on tyrosine protein kinase inhibitors are the most common treatments. The latter group have been used as the primary choice for a decade. However, tumor microenvironment in HCC is strongly immunosuppressive; thus, new treatment approaches for HCC remain necessary. The great expression of immune checkpoint molecules, such as programmed death-1 (PD-1), cytotoxic T-lymphocyte antigen 4 (CTLA-4), lymphocyte activating gene 3 protein (LAG-3), and mucin domain molecule 3 (TIM-3), on tumor and immune cells and the high levels of immunosuppressive cytokines induce T cell inhibition and represent one of the major mechanisms of HCC immune escape. Recently, immunotherapy based on the use of immune checkpoint inhibitors (ICIs), as single agents or in combination with kinase inhibitors, anti-angiogenic drugs, chemotherapeutic agents, and locoregional therapies, offers great promise in the treatment of HCC (please refer to PMID: 34065489 and expand)

-> Thank you for your valuable comment. We inserted the mentioned article as a reference[11] in the introduction section, lines 68-70.

-> Hepatocellular carcinoma (HCC) is the most common type of primary liver cancer and the most common cause of death in people with cirrhosis[10]. The prognosis for HCC is poor because it tends to be diagnosed at an advanced stage. It is often asymptomatic until the tumor becomes large enough to tighten the liver capsule or compress adjacent organs or vessels with nerves, causing pain. The other cause of poor prognosis is that the state of the underlying liver disease limits treatment options and has an adverse effect on results regardless of the HCC stage. Due to the immunosuppressive tumor microenvironment of HCC, the new treatment targets and strategies have been proposed[11], and the new biomarker for predicting prognosis is necessary.

Round 2

Reviewer 1 Report

The authors have replied almost all the questions.

Reviewer 2 Report

The authors have clarified several of the questions I raised in my previous review. Most of the major problems have been addressed by this revision.